# Improved Pancreatic Cancer Detection and Localization on CT Scans: A Computer-Aided Detection Model Utilizing Secondary Features

**DOI:** 10.3390/cancers16132403

**Published:** 2024-06-29

**Authors:** Mark Ramaekers, Christiaan G. A. Viviers, Terese A. E. Hellström, Lotte J. S. Ewals, Nick Tasios, Igor Jacobs, Joost Nederend, Fons van der Sommen, Misha D. P. Luyer

**Affiliations:** 1Department of Surgery, Catharina Cancer Institute, Catharina Hospital Eindhoven, EJ 5623 Eindhoven, The Netherlands; misha.luyer@catharinaziekenhuis.nl; 2Department of Electrical Engineering, Eindhoven University of Technology, AZ 5612 Eindhoven, The Netherlands; c.g.a.viviers@tue.nl (C.G.A.V.); t.a.e.hellstrom@tue.nl (T.A.E.H.); fvdsommen@tue.nl (F.v.d.S.); 3Department of Radiology, Catharina Cancer Institute, Catharina Hospital Eindhoven, EJ 5623 Eindhoven, The Netherlands; lotte.ewals@catharinaziekenhuis.nl (L.J.S.E.); joost.nederend@catharinaziekenhuis.nl (J.N.); 4Department of Hospital Services and Informatics, Philips Research, AE 5656 Eindhoven, The Netherlands (I.J.)

**Keywords:** pancreatic ductal adenocarcinoma, early detection, deep learning, artificial intelligence, computer-aided detection, computed tomography, secondary features

## Abstract

**Simple Summary:**

Pancreatic ductal adenocarcinoma (PDAC) is one of the most aggressive cancers, and most patients present with advanced or irresectable disease due to late recognition. Radiological imaging modalities such as CT scans are key in providing information on the presence or absence of tumors. However, an assessment of pancreatic cancer requires specific radiological expertise, and small tumors are easily overlooked. Computer-aided detection (CAD) using artificial intelligence (AI) techniques is promising and may help in the early detection of pancreatic tumors. In this study, we developed a deep learning-based tumor detection framework that can detect pancreatic head cancer on CT scans with high accuracy when incorporating clinically relevant information. We demonstrate that a tumor detection framework utilizing CT scans and secondary signs of pancreatic tumors results in an increased detection accuracy for the detection of pancreatic head tumors.

**Abstract:**

The early detection of pancreatic ductal adenocarcinoma (PDAC) is essential for optimal treatment of pancreatic cancer patients. We propose a tumor detection framework to improve the detection of pancreatic head tumors on CT scans. In this retrospective research study, CT images of 99 patients with pancreatic head cancer and 98 control cases from the Catharina Hospital Eindhoven were collected. A multi-stage 3D U-Net-based approach was used for PDAC detection including clinically significant secondary features such as pancreatic duct and common bile duct dilation. The developed algorithm was evaluated using a local test set comprising 59 CT scans. The model was externally validated in 28 pancreatic cancer cases of a publicly available medical decathlon dataset. The tumor detection framework achieved a sensitivity of 0.97 and a specificity of 1.00, with an area under the receiver operating curve (AUROC) of 0.99, in detecting pancreatic head cancer in the local test set. In the external test set, we obtained similar results, with a sensitivity of 1.00. The model provided the tumor location with acceptable accuracy obtaining a DICE Similarity Coefficient (DSC) of 0.37. This study shows that a tumor detection framework utilizing CT scans and secondary signs of pancreatic cancer can detect pancreatic tumors with high accuracy.

## 1. Introduction

Pancreatic ductal adenocarcinoma (PDAC) has a dismal prognosis and a poor overall 5-year survival rate of only 9% [1,2]. Pancreatic cancer is one of the most aggressive malignancies and often goes undetected until it has progressed into an advanced stage. As a result, most patients present with advanced or irresectable disease, leading to limited treatment options and poor outcomes [3,4]. The initial diagnosis of pancreatic cancer usually starts with performing a CT scan [5]. However, the assessment of pancreatic cancer requires specific radiological expertise, and small tumors are easily overlooked, especially in asymptomatic patients [6]. Lack of expertise may result in delayed recognition, depriving patients of potentially curative treatment [7]. Since pancreatic cancer treatment is centralized, this may be even more apparent in hospitals without specific pancreatic expertise [8]. Radiologists’ sensitivity of detecting small and isoattenuating PDACs with sizes smaller than 2 cm on CT has been reported to be between 58% and 77% [9]. In addition, a recent study demonstrated that indicative changes associated with PDAC are often visible on imaging 6–18 months prior to actual diagnosis [10]. However, it is estimated that in the months prior to the clinical diagnosis of PDAC, the sensitivity of CT drops to 44%.

Computer-aided detection (CAD) using artificial intelligence (AI) techniques is promising and may help in the early detection of pancreatic tumors [11]. Deep learning-based algorithms can provide pixel-level segmentation of relevant anatomy. Deep learning refers to the use of a neural network with multiple layers, suited for recognizing features from input data. Deep learning using convolutional neural networks (CNNs) has rapidly advanced image processing and, as a result, the development of image-based CAD methods [12]. These models can integrate expert knowledge into their operation, providing significant value during disease screening, particularly in non-expert centers. However, for successful clinical implementation, the algorithm must provide interpretable outcomes that reveal its functioning. The aim of this study is to investigate the potential of a deep learning-based algorithm for the detection of pancreatic tumors. Acknowledging the value of secondary cancer-indicative features on CT, the algorithm incorporates these features to improve detection performance. This study presents a deep learning-based framework for detecting pancreatic head cancer on CT scans with high accuracy by incorporating clinically relevant information. The results show that including secondary signs of pancreatic tumors enhances detection accuracy and allows for automated segmentation of tumors and relevant anatomy. The framework’s performance is validated using both an internal test set and a publicly available test set, achieving comparable results.

## 2. Materials and Methods

This study was approved by the Medical Ethical Board of the Catharina Hospital Eindhoven (CZE) under registration number W19.251 and the Internal Committee of Biomedical Experiments of Philips. Informed consent from all patients was waived due to the dismal prognosis of pancreatic cancer and the retrospective design of this research. Before model training and testing, all data in this study were pseudonymized.

### 2.1. Data Collection and Labelling

We retrospectively collected CT images of 98 control patients and 99 patients with pathology-proven adenocarcinoma in the pancreatic head. The PDAC cohort consisted of patients aged 18 years and above who underwent treatment for resectable or borderline resectable pancreatic head cancer at the Catharina Hospital Eindhoven between 2012 and 2019. Patients were eligible for inclusion if their CT images included at least two phases and were accompanied by both a surgical report and a complete pathology report. Exclusion criteria included patients diagnosed with active pancreatitis at the time of diagnosis or those with artifacts on CT images, such as metal stents. The control group consisted of patients from the randomized NUTRIENT-II trial, involving patients with esophageal cancer but not with pancreatic tumors [13]. All patients underwent a preoperative CT scan as part of their diagnostic workup. The dataset comprised medical imaging data and relevant clinical information, including pre-, intra-, and postoperative details obtained from radiology, pathology, and surgery reports. The medical imaging data consisted of CT scans utilizing different phases. A total of 198 CT scans were included in the PDAC cohort. The included CT scans consisted of a portal venous phase, parenchymal phase, or (late) arterial phase, allowing for the integration of complementary information when obtaining a patient-level cancer prediction. The CT slice thickness varied between 1.0 and 3.0 mm based on accessibility while also aiming to integrate the diversity of scans encountered in clinical practice. A medical doctor (M.R.) manually annotated relevant anatomical structures, consisting of the tumor, pancreas, pancreatic duct, and common bile duct. Bile and pancreatic ducts were annotated in both groups if visible. Annotations were performed using IntelliSpace Portal (Philips, The Netherlands, Eindhoven) and were supervised by an expert radiologist in the field of pancreatic tumors (J.N.).

Additionally, we used the publicly available Medical Segmentation Decathlon (MSD) dataset from the Memorial Sloan Kettering Cancer Center (Manhattan, NY, USA) for model testing [14]. This dataset consists of abdominal CT scans of 281 patients with mostly advanced and diverse pancreatic malignancies (intraductal papillary neoplasms, pancreatic neuroendocrine tumors, or PDAC). In this dataset, portal-venous phase CT scans were used with varying slice thickness. To evaluate the performance of our model in comparison to other existing methods on a recognized benchmark, we re-annotated 10% of the dataset (28 cases) with suspected adenocarcinoma in the pancreatic head with a level of detail similar to our dataset. This subset served as an additional unseen test set during our experimental evaluations, providing insights into the efficacy of our model.

### 2.2. PDAC Segmentation for Classification Framework

We developed an AI system designed to detect pancreatic cancer, aiding clinicians in early diagnosis with clinically interpretable results. Therefore, we made use of secondary signs (such as duct dilatation) to enhance interpretability. A “multi-stage coarse-to-fine” framework was developed that divides the detection process into multiple steps, where each step progressively refines the detection accuracy, starting from a coarse level of analysis and gradually refining to more detail. This approach consists of (1) an initial pancreas localization model, (2) a fine pancreas segmentation model, (3) a common bile duct and pancreatic duct segmentation model, and (4) a final tumor detection model that employs these predicted features in the tumor segmentation algorithm. First, the coarse segmentation of the pancreas was carried out in a downsampled full-image CT scan. The segmentation of the pancreas was accomplished using a coarse-to-fine 3D U-Net-based approach. Initially, a coarse model processed full-image CT scans, which were resampled to twice (1.37 mm, 1.37 mm, and 2 mm) the target resolution (0.68 mm, 0.68 mm, and 1 mm). This model employed a patch-based strategy, utilizing patches sized at 128 × 128 × 64 voxels with a stride of 32 × 32 × 32 voxels. Subsequently, a fine segmentation model was applied to a high-resolution, cropped region (256 × 256 × 192 voxels) around the center of the pancreas, as determined from the coarse segmentation. The fine model, also based on a U-Net architecture, focused specifically on these cropped regions to enhance the precision of pancreatic segmentation. It employed the same patch size of 128 × 128 × 64 voxels and a stride of 32 × 32 × 32 voxels. Both the coarse and fine models consisted of a four-layer deep U-Net, with each layer containing 32, 64, 128, and 256 convolutional filters, respectively. However, they were trained on data of different resolutions to optimize their performance at the stage in the processing chain. The ductal segmentation models were also applied to the high-resolution crop around the pancreas. These segmentation models similarly used a patch-based 3D U-Net model trained on the cropped region of the CT scan. We used a 3D U-Net-like deep CNN due to its specific applicability to medical segmentation [15]. The models segmented patches of size 128 × 128 × 128 pixels at a stride of 32 × 32 × 32 pixels. The tumor segmentation model took the full CT crop (256 × 256 × 192) along with the segmentations of the previously extracted anatomical features as input. Thus, the model had a large receptive field, enabling it to consider aspects such as the ductal size (potential dilation) in the context of the pancreas and the CT crop, allowing for maximally informed tumor predictions. Since we provided the segmentation of the pancreas and ducts as input to the detection model, the model was trained to utilize any valuable feature from these segmentations. Therefore, the model potentially also included other secondary features such as atrophic pancreas, ductal interruption, and peri-pancreatic infiltration. The complete workflow is shown in Figure 1 and Figure 2.

Two different methods were used for bile duct and pancreatic duct segmentation: (1) A multi-label model was developed that segmented both bile duct structures simultaneously, and (2) two separate single-label models were employed to individually segment each structure [16]. Additionally, to further reinforce realistic segmentations, bile duct-connected components that did not originate from pancreas segmentation were categorized as background using the connected components 3D (cc3d) package.

Data preprocessing for training and at test time involved resampling the CT scans to both low-resolution (1.37 mm, 1.37 mm, and 2 mm) and high-resolution (0.68 mm, 0.68 mm, and 1 mm) voxel spacing, in addition to normalizing the CT scans to a range of −87 to 199 Hounsfield units. Variously sized volumes were processed at different stages of the detection pipeline; refer to Figure 1 for a depiction of these sizes. During training various data augmentation techniques were employed. Data augmentation for the pancreas and secondary feature models primarily included random rotations and elastic deformations. For the tumor detection model, we utilized an extended series of augmentations, namely flipping, rotating, eroding, and dilating the provided ducts, along with elastic deformation, additive Gaussian noise, and additive Poisson noise. These augmentations enhanced the model’s robustness and ability to generalize across different imaging conditions and patient anatomies.

All models were trained using a threefold bootstrapping method with 70% of the internal dataset (per patient) used to train the model and 15% for validation. This approach led to three distinct models for each step of the detection process, all acting as a separate opinion on the specific task. The approach for tumor detection integrated a series of models in an ensemble framework. First, the cropped pancreas was fed into three pancreas segmentation models trained on the different bootstrapped training subsets. These refined segmentation procedures yielded (2) three precise, pancreas-centered crops, each uniquely tailored to capture the organ’s intricate details. Subsequently, these crops were processed through (3) respective secondary feature models for ductal segmentation, specifically designed for each fold. This step was crucial for generating three accurate sets of ductal segmentations. The next stage involved stacking these segmentations with the original CT crop, forming the input for the (4) three tumor segmentation models. From this, three separate tumor predictions were derived, each representing a different perspective of the potential tumor. Finally, (5) a combination of the three models’ predictions was collated as the final PDAC classification and segmentation prediction [17]. This ensemble consists of multiple individual models, each trained on different subsets of data (cross-validation folds) and with different initialization parameters. By aggregating the predictions from these diverse models (mean of the predicted probabilities), we can achieve more accurate and reliable combined results compared to using a single model alone. This strategy not only helps to mitigate the biases and variances inherent in individual models but also ensures that the final output is more robust to variations in the input data. This layered and iterative process not only maximizes the precision of tumor detection but also leverages the strengths of multiple models to provide a more comprehensive and reliable diagnosis. To test the approach, we applied the complete detection workflow to a predefined representative test set encompassing 15% of our data. In addition, the models were applied to the public MSD dataset as a separate external test set.

### 2.3. Statistical Analysis

We performed statistical analyses using the Statistical Package for the Social Sciences (SPSS, version 29). Descriptive statistics were calculated for demographic variables, including mean and corresponding standard deviation, median including interquartile range, and frequency for categorical variables.

We assessed our model’s performance using two methods. (1) localization performance: We identified true-positive cases if the predicted segmentation overlaps with the ground-truth tumor, evaluating tumor detection capability. (2) classification accuracy: We compared the voxel with the highest prediction value in the tumor prediction map to the actual tumor presence, quantifying performance via AUROC curves. The area under the curve (AUC) was calculated to assess the overall performance of the tumor detection model. Case-level performance evaluation was conducted by constructing receiver operating characteristic (ROC) curves and determining AUC values, reflecting the model’s ability to differentiate between tumor-positive and tumor-negative cases. ROC curve analysis allowed for the assessment of the model’s discrimination between tumors and non-tumors across different thresholds. Additionally, individual model performance within the proposed sequential processing workflow was assessed by determining sensitivity and specificity at an optimal threshold of 0.61, established on the validation set, and/or segmentation accuracy measured using the DICE Similarity Coefficient (DSC). The DSC represents the degree of spatial overlap between the predicted and the ground-truth segmentation. Since we used a segmentation-for-detection model, sensitivity represents the ratio of correctly identified positive cases where the tumor was predicted and confirmed through ground-truth segmentation. The specificity quantifies the proportion of true-negative cases with no AI-segmented tumor and no corresponding ground-truth tumor segmentation.

## 3. Results

### 3.1. Clinical Characteristics

The dataset contained 197 patients with 290 CT volumes. A total of 99 patients were diagnosed with pancreatic head cancer, corresponding with 198 CT volumes. The remaining 98 patients were assigned to the control group with a normal pancreas. The clinical characteristics of the patients, classified by the presence or absence of PDAC, are shown in Table 1. The pancreatic cancer cohort consisted of 52 male and 47 female patients, who had a mean age of 74.9 ± 7.5 years. The control cohort consisted of 79 male and 18 female patients, with a mean age of 71.2 ± 8.1 years. In total, 21 patients had stage I pancreatic cancer, 55 patients had stage II, 20 patients had stage III, and 3 patients had stage IV pancreatic cancer. A total of 77 patients presented with hypoattenuating tumors, 21 with isoattenuating tumors, and 8 with hyperattenuating tumors. Additionally, 52 patients had pancreatic carcinoma, 21 had cholangiocarcinoma, and 24 had ampullary carcinoma. The medium tumor size was 2.6 cm (range 2.0–3.5).

### 3.2. Segmentation of Secondary Features

The fine pancreas segmentation models achieved mean DSCs of 0.86 ± 0.05 on the internal test set and 0.88 ± 0.03 on the MSD dataset (Table 2). For bile duct segmentation, a multi-class model achieved mean DSCs of 0.61 ± 0.22 for the common bile duct and 0.49 ± 0.25 for the pancreatic duct on our local test set. It achieved mean DSCs of 0.67 ± 0.19 and 0.51 ± 0.19 for the common bile duct and pancreatic duct, respectively, on the MSD dataset. The models performed consistently between the internal test set and the MSD dataset.

### 3.3. Performance of Tumor Detection Model

All models were applied to the internal test set and the MSD dataset as test sets. Utilizing ground-truth annotations of the pancreas and ducts as input, the tumor detection model demonstrated perfect accuracy in identifying tumors within the internal test set, resulting in a sensitivity of 1.00 (Table 3). The tumor detection model achieved a specificity of 0.86 on the internal test set. Combining the three different models improved performance with no false-positive predictions, resulting in a perfect 1.00 sensitivity and specificity. Utilizing the multi-stage algorithm where the pancreas, common bile duct, and pancreatic duct were automatically segmented and provided to the tumor detection model, the specificity dropped to 0.64 ± 0.15. However, combining the different models improved the overall performance to a sensitivity of 0.97 (1 case missed) and a specificity of 1.0. Overall, the models displayed accurate tumor detection results, with an AUROC of 0.99. In a subanalysis of tumors smaller than 2 cm within the internal test set (12 cases < 2 cm vs. control set), the model achieved an impressive AUROC of 0.98. However, it also recorded a much lower DSC of 0.19 ± 0.24, which highlights the challenge of accurately delineating these small tumors (Figure 3).

Using the ground-truth annotations of relevant anatomy and the multi-stage algorithm as inputs, the tumor detection model demonstrated a perfect sensitivity of 1.00 within the MSD test set in both instances (Table 4). The model segmented the tumor at a mean DSC of 0.37 in both the internal test set and the MSD test set.

Finally, we visualized the predicted segmentations generated by the multi-stage algorithm in the internal test set to understand which aspects contributed to the algorithm’s performance (Figure 4).

## 4. Discussion

This study shows accurate pancreatic tumor detection using a deep learning framework employing multi-stage algorithms on contrast-enhanced CT scans. It underscores the significance of integrating clinically relevant information in the development of CAD methods, facilitating interpretation by clinicians in real-world clinical practice.

Recent studies have applied classification networks to achieve the detection of PDAC and other pancreatic tumors on CT scans, with particular attention to segmentation-based classification, which contributes to both cancer detection and tumor localization [18,19,20,21,22,23,24]. Additionally, Viviers et al. previously compared several approaches for improving deep learning-based pancreatic tumor detection, ultimately showcasing the value in clinically relevant and tumor-indicative secondary features in the low-data regime [18]. The literature indicates that 5.4–14% of pancreatic tumors present as completely isoattenuating, making them indistinguishable from normal pancreatic tissue [25]. Particularly in such cases, radiologists rely on alternative patterns suggestive of malignant disease. Over time, multiple studies have consistently reported the presence of visible secondary features, prior to actual diagnosis [26,27]. For instance, Kang et al. found that 88% of the cases exhibited such secondary signs [28]. Additionally, imaging has revealed indicative changes associated with PDAC up to 18 months before diagnosis in 50% of patients [10]. These findings confirm that incorporating secondary features of pancreatic cancer improves pancreatic tumor detection [6].

Although the prediction of the pancreas segmentation algorithm presented in this study was irregular around the edges of the pancreas, the performance of this model was comparable with the current state-of-the-art pancreas segmentation models [29,30]. For instance, Huang et al. introduced a semiautomated DUNet aiming at capturing the irregular shape variations of the pancreas, thereby enhancing the accuracy of pancreas segmentations, resulting in a mean DSC of 87.25 ± 3.27 [30]. Although their approach resulted in a high segmentation accuracy, they also seemed to suffer from the large shape variations and irregular boundaries of the pancreas.

Given the pivotal diagnostic information provided by adjacent structures such as a dilated pancreatic duct and common bile duct, the framework incorporated this information to establish an understanding of the relationship between tumors and ducts, with a special focus on the bile duct. The algorithm achieved high sensitivity in identifying all bile ducts but had slightly lower specificity due to false-positive predictions. The smaller pancreatic duct size led to decreased DSC values. Variations in specificity and sensitivity measures on the test set were notable, largely due to occasional predictions of ducts without ground-truth annotations. However, mispredictions commonly occur in areas where bile ducts are typically found, implying potential validity. While the results of this study are encouraging, further refinement is crucial for more accurate predictions and broader model applicability, as well as to potentially reduce false-positive predictions [31].

In this study, by employing ground-truth annotations and a multi-stage algorithm, the tumor detection model showcased exceptional sensitivity. It achieved a perfect sensitivity score of 1.00 within the MSD test set for both cases. With an AUROC of 0.99, our models improve the current state-of-the-art approaches, particularly in external validation on the MSD dataset. Although multiple research approaches utilized classification networks, only a few studies have externally validated their models, with two studies utilizing the publicly accessible MSD dataset [19,32]. Alves et al. introduced an automated framework for PDAC detection, employing three nnUnet models and incorporating secondary features for tumor presence evaluation [19]. They demonstrated the benefit of integrating anatomical information, and despite not presenting sensitivity and specificity outcomes, they reported a notable AUC-ROC of 0.91. Liu et al. developed a deep learning model utilizing a modified VGG network to differentiate between pancreatic cancer tissue and non-cancerous tissue [32]. Their model was also evaluated using the MSD dataset, achieving a sensitivity of 0.79 and specificity of 0.84, resulting in a maximum AUC-ROC of 0.92 at the patient level. The results in this study highlight the importance of incorporating secondary tumor-indicative information and represent a notable improvement in pancreatic tumor detection accuracy. Despite achieving a mean DSC of 0.34, indicating limited tumor localization accuracy, our primary focus was the identification of tumor presence rather than delineation, given the significant benefits of early detection for patient survival [11,33].

Recent studies have applied classification networks to detect PDAC and other pancreatic tumors, marking an important first step in enhancing pancreatic cancer diagnosis [18,19,20,21,22,23,24]. However, high variability in resectability assessments can compromise treatment recommendations [34]. Therefore, future research should focus on improving resectability assessments. Notably, the study by Viviers et al. is one of the first to propose a framework for assessing the resectability of pancreatic cancer [35]. This approach aims to improve accuracy and reduce variability by analyzing the anatomical relationships between the tumor and surrounding vasculature. Continued research should further refine these assessments to enhance treatment reliability.

This study has several limitations. First, the models were trained using a relatively small dataset, and we exclusively focused on discriminating pancreatic head cancer from normal pancreas CT scans. The model was not subjected to images displaying other neoplastic or inflammatory pathologies or tumors located in different regions of the pancreas. Future research will include a more diverse dataset to encompass various conditions to enhance the framework’s ability to differentiate between these conditions. Secondly, the dataset originated from a specific cohort with an uneven distribution of cancerous and non-cancerous cases. Moreover, we utilized various scan phases driven by accessibility considerations. Therefore, the dataset may not reflect real-world prevalence. This could have led to distribution bias, and therefore, a prospective study is needed to evaluate our models’ performance on real-world clinical data. These studies should use large, multicenter datasets and include a more diverse patient population including inflammatory and benign pancreatic lesions to ensure the model’s robustness and generalizability across different demographics. Finally, the tumor detection models were externally validated using the MSD dataset. Although this dataset is valuable, it mainly consists of patients with late-stage disease. As a result, this dataset contains a high proportion of large tumors and metal stents. Since we used a subselection of the total dataset, this could have induced bias in the evaluation of our models. Furthermore, since the MSD dataset exclusively comprises tumor-positive cases, specificity was not assessed in this study.

## 5. Conclusions

This study demonstrates that a tumor detection framework utilizing CT scans and secondary signs of pancreatic tumors as input results in an increased sensitivity and specificity for detecting pancreatic head tumors. The multi-stage detection models integrate detailed duct and pancreas segmentation maps with CT scans, enabling them to establish connections between these secondary features and tumor presence. Furthermore, this study provides valuable insights into the model’s performance, addressing concerns regarding result explainability and occasional unexpected recommendations. Employing AI algorithms for early pancreatic cancer detection has the potential to increase the likelihood for patients to be eligible for curative treatment, thereby potentially enhancing survival outcomes. Despite these encouraging results, further research utilizing large multicenter datasets is necessary to validate the outcomes observed in this study.

## Figures and Tables

**Figure 1 cancers-16-02403-f001:**
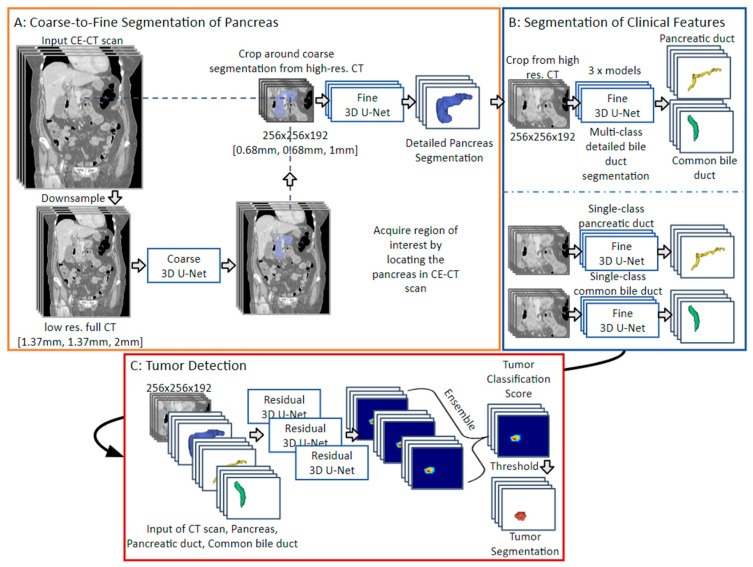
Flow diagram of the multi-stage framework of AI models for pancreatic tumor detection. The coarse-to-fine pancreas segmentation process is depicted in Block (**A**) (Orange). Block (**B**) (Blue) depicts multiple methods to segment clinical features. In Block (**C**) (Red), the PDAC segmentation process is depicted from which the final tumor score is obtained.

**Figure 2 cancers-16-02403-f002:**
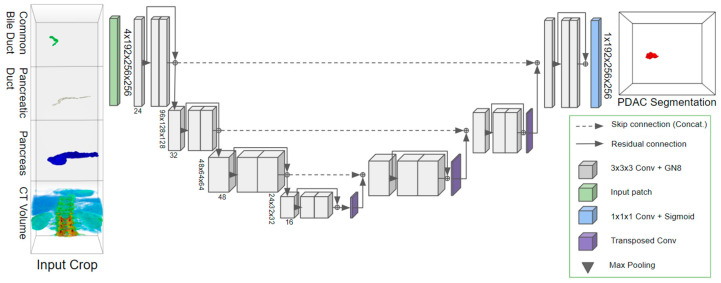
Diagram of the residual 3D U-Net utilizing secondary features for improved segmentation of pancreatic tumors.

**Figure 3 cancers-16-02403-f003:**
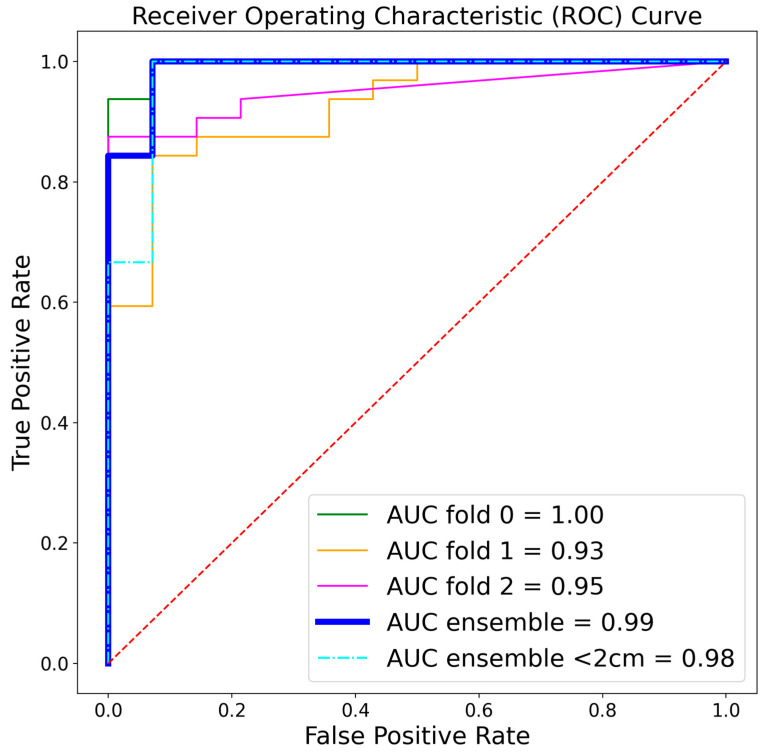
Receiver operating characteristic (ROC) curves of the separate folds and the full tumor detection approach (ensemble) on the internal test set. AUC = area under the curve.

**Figure 4 cancers-16-02403-f004:**
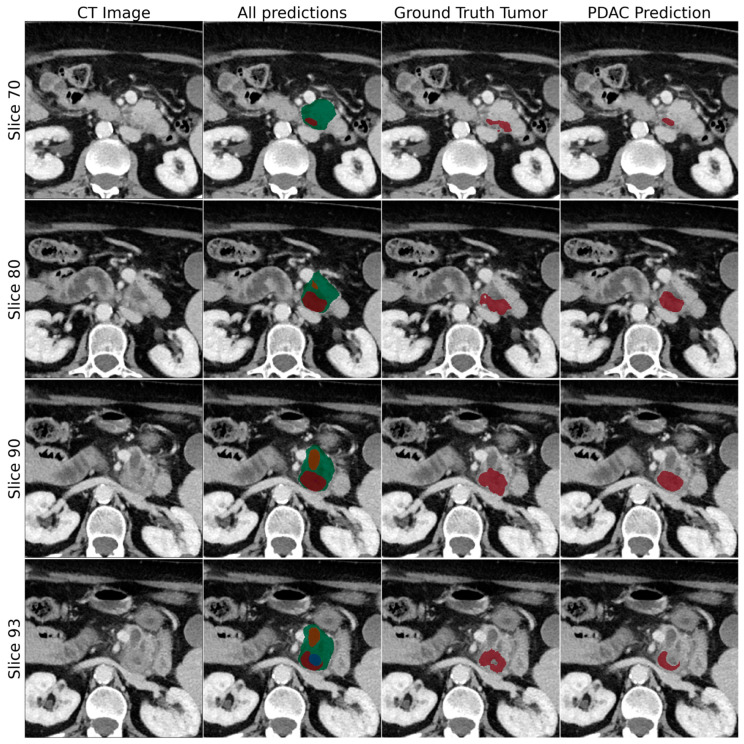
This figure shows examples of CT images, included with predicted segmentations of the proposed tumor detection framework. CT scans are in the parenchyma phase. The tumor is depicted in red, the pancreas in green, the pancreatic duct in orange, and the common bile duct in blue. The ground-truth tumor and tumor prediction are depicted in consecutive columns. In this case, the model achieved a DICE score of 0.71 on an image from the test set.

**Table 1 cancers-16-02403-t001:** Clinical characteristics of the patients in the pancreatic cancer cohort and control cohort.

Clinical Characteristics	With Pancreatic Head Cancer	Without Pancreatic Cancer
N =	99	98
Age (years)	74.9 ± 7.5	71.2 ± 8.1
Gender (M/F)	52/47	79/19
Tumor stage		
I/II/III/IV	21/55/20/3	N.A.
Tumor attenuation on CT		
Hypo/Iso/Hyper intens	77/14/8	N.A.
Tumor origin		
Pancreas/Cholangio/Ampullary	54/21/24	N.A.
Tumor size (cm)	2.60 (2.0–3.5)	N.A.

Continuous variables are displayed as mean ± standard deviation or median (interquartile range). The tumor stages are I: T1-2N0 pancreatic cancer; II: T3 or T1-3N1 pancreatic cancer; III: T4 or T1-3N2 pancreatic cancer; IV: metastasized. Tumor stages and tumor size are only presented for pancreatic cancer patients. N.A. = not applicable.

**Table 2 cancers-16-02403-t002:** Results of the pancreas and multi-class bile duct segmentation models.

	Internal Test Set	Medical Decathlon Dataset
Pancreas	0.86 ± 0.05	0.88 ± 0.03
Common bile duct	0.61 ± 0.22	0.67 ± 0.19
Pancreatic duct	0.49 ± 0.25	0.52 ± 0.19
Common bile duct *(Excluding not annotated)*	0.63 ± 0.18	0.69 ± 0.14
Pancreatic duct *(Excluding not annotated)*	0.60 ± 0.11	0.55 ± 0.13

Results are evaluated using the DICE Similarity Coefficient (DSC) for the internal test set and the Medical Segmentation Decathlon (MSD) dataset.

**Table 3 cancers-16-02403-t003:** Tumor detection model results for pancreatic cancer detection on the internal test set.

	GT Input	GT Input—Ensemble	Predicted Input	Predicted Input—Ensemble (ALL)	Predicted Input—Ensemble (<2 cm)
Sensitivity	1.00 ± 0.00	1.00	1.00 ± 0.00	0.97	1.0
Specificity	0.86 ± 0.10	1.00	0.64 ± 0.15	1.00	1.0
Precision	0.94 ± 0.04	1.00	0.87 ± 0.05	1.00	1.0
F1	0.97 ± 0.02	1.00	0.93 ± 0.03	0.98	1.0
Accuracy	0.96 ± 0.03	1.00	0.89 ± 0.05	0.98	1.0
ROC	0.96 ± 0.02	0.98	0.97 ± 0.03	0.99	0.98
Mean Dice	0.35 ± 0.04	0.37	0.31 ± 0.02	0.34	0.190 ± 0.24

Evaluation using the DICE Similarity Coefficient (DSC), sensitivity, and specificity on the internal test dataset was performed. Methods were compared using manually annotated input of the pancreas and bile ducts and employing the automatically segmented bile ducts and pancreas using AI models. Ensemble refers to the aggregation of predictions from three differently trained tumor prediction models (details in Section 2.2). GT = ground truth. F1 = calculated as the mean of the precision and recall scores. ROC = receiver operating characteristic.

**Table 4 cancers-16-02403-t004:** Tumor detection model results on the Medical Segmentation Decathlon dataset.

	GT Input	GT Input—Ensemble	Predicted Input	Predicted Input—Ensemble
Sensitivity	1.00 ± 0.00	1.00	1.00 ± 0.00	1.00
Mean Dice	0.37 ± 0.03	0.37	0.37 ± 0.01	0.37

Evaluation using the DICE Similarity Coefficient (DSC) and sensitivity on the Medical Segmentation Decathlon (MSD) dataset was performed. Methods were compared using manually annotated input of the pancreas and bile ducts and employing the automatically segmented bile ducts and pancreas using AI models. GT = ground truth.

## Data Availability

The code for the models presented in this study is available via https://github.com/cviviers/3D_UNetSecondaryFeatures/tree/all_models (accessed on 1 June 2024).

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
