# Peer review of "Improved Pancreatic Cancer Detection and Localization on CT Scans: A Computer-Aided Detection Model Utilizing Secondary Features"

_cancers, 2024, doi:10.3390/cancers16132403_

Round 1

Reviewer 1 Report

Comments and Suggestions for Authors

The authors address an important issue, as in 90 % pancreatic tumors are detected too late. The study shows the feasibility of KI supported diagnosing. 

my questions/issues: 

please add data on the detection rate by radiologists for your cases. It is important to aid in small and difficult to detect tumors.

Please include tumors smaller than 2 cm

From the clinical point of view: Please specify the ability of your tool to predict the resectability of the tumor

I feel a perfect sensitivity and specifity won't persist in a real world situation.  Where do you exspect difficulties for your model to detect tumors? Which weaknesses do you see? Did you compare with bening pacreatic lesions, e.g. mass forming pancreatitis, inflammatory pseutoumors etc.?

Author Response

Thank you for your comments. Attached you will find the point-by-point reply. 

Reviewer 2 Report

Comments and Suggestions for Authors

The authors propose a multi-stage coarse-to-fine framework for pancreatic cancer detection and localization based on CT images. The framework includes a pancreas localization model, a fine pancreas segmentation model, a common bile duct and pancreatic duct segmentation model and a final tumor detection model. U-Net is applied as the base deep learning framework in those models. Experiments were done on two datasets, the results seem satisfying.

Major comments:

1. The introduction chapter does not clearly state the contribution of the manuscript.

2. The Coarse 3D U-Net and Fine 3D U-Net are not described clearly.

3. Due to the lack of comparison with existing methods, the results in this paper are not convincing enough.

4. Missing details on the images in the data sets, e.g., image size (x, y, and z)? tumor size range (in pixel)? any preprocessing techniques applied? Any data augmentation applied?

Minor comments:

1. Red wave underlines exist in Figure 1.

2. Table 3 needs to specify clearly what does ‘Ensemble’ mean.

Author Response

(The authors gave the same response as above.)

Reviewer 3 Report

Comments and Suggestions for Authors

Dear Editor and Authors,

I have read the manuscript carefully and found no substantial errors. The language is appropriate for a scientific article, the references are recent and relevant, and the figures are well done. The results obtained support the conclusions well. The manuscript certainly deserves publication.

I only found two points that require revision. I suggest accepting the manuscript after minor revisions.

1. The criterion for choosing patients is not very clear.

2. The authors write, "Despite these encouraging results, further research utilizing large multicenter datasets is necessary to validate the outcomes observed in this study." What kinds of further studies do the authors suggest?

Additionally, please review the entire manuscript for typographical errors.

Comments on the Quality of English Language

Good

Author Response

(The authors gave the same response as above.)

Reviewer 4 Report

Comments and Suggestions for Authors

Ramaekers  et al. conducted a retrospective study by collecting CT images from 99 patients with pancreatic ductal adenocarcinoma (PDAC) and 98 control subjects. They developed a deep learning-based framework for tumor detection that identified pancreatic head cancer on CT scans with high accuracy when incorporating secondary signs, such as dilation of the pancreatic duct and common bile duct.

It is notable that Ref 10 by Singh demonstrated that indicative changes associated with PDAC are often visible on imaging 6 to 18 months prior to actual diagnosis. Singh's study utilized CT scans obtained 3 to 36 months before the diagnosis of PDAC. In contrast, the current study used CT scans taken at the time of diagnosis, despite its aim to facilitate early detection of PDAC. The authors enhanced interpretability by incorporating secondary signs like duct dilation. However, it remains unclear whether other secondary signs were used. The authors should specify if additional secondary signs were included.

The median tumor size reported was 2.6 cm. It is crucial to confirm whether low-density pancreatic masses/hypodensity were included as primary signs in the model, and whether any of the tumors were isoattenuated. Additionally, the inclusion of vascular involvement in the model should be clarified.

The gender distribution in the PDAC cohort versus the control cohort was male/female 53%/47% versus 81%/19%, respectively. May this discrepancy in gender distribution  have impacted the results? For comparison, Ref 10 matched the cases and controls based on both age and gender.

In the discussion, the authors mention that the high proportion of large tumors and the presence of metal stents could have introduced bias in the evaluation of the models. However, the number of patients with a metal stent was not specified. Additionally, it is important to address the feasibility of using CT scans from patients with metal stents in situ, as duct dilation can be affected by the presence of a stent.

Figures 1 and 2 are difficult to read as presented and should be enlarged for better clarity.

Comments on the Quality of English Language

No comments.

Author Response

(The authors gave the same response as above.)

Round 2

Reviewer 2 Report

Comments and Suggestions for Authors

The authors propose a multi-stage coarse-to-fine framework for pancreatic cancer detection and localization based on CT images. The framework includes a pancreas localization model, a fine pancreas segmentation model, a common bile duct and pancreatic duct segmentation model and a final tumor detection model. U-Net is applied as the base deep learning framework in those models. The proposed method obtained satisfying performance on the testing dataset.

Author Response

Dear sir/madam, 

Since no additional revisions were suggested no adjustments were made to the manuscript. 

Reviewer 4 Report

Comments and Suggestions for Authors

The authors have addressed all my comments in my first review satisfactory.

I have only one minor comment. From table 1 there were 57 patients with pancreatic cancer,  21 with distal cholangiocarcinoma and 24 with ampullary carcinoma. First, the correct headline for this column should be "Pancreatic head cancer" and not "Pancreatic cancer", and the title of Table 1 should not use the term PDAC. Second, Results line 235, the "PDAC cohort" is not correct, since only 57 patients had PDAC.  PDAC=Pancreatic ductal adenocarcinoma and do not include cholangiocarcinoma or ampullary cancer. Third, Abstract line 31,  "pancreatic cancer" should be "pancreatic head cancer". From a clinical point of view this distinction is important. Be consistent troughout the manuscript. Would the authors comment on this in the Introduction or Discussion, i.e. that "Pancreatic head cancer" include four different histological entities: pancreatic ductal adenocarcinoma (PDAC), distal cholangiocarcinoma (dCCA), duodenal adenocarcinoma (DAC), and ampullary carcinoma. Please crf Uijterwerk et al, Ann Surg Oncol 2024, 10.1245/s10434-024-15555-8 

Author Response

Dear sir/madam, 

Thank you for your additional comments. In the attached file you will find a point-by-point reply regarding your comments. 

Kind regards, 
